# Oleate Impacts on Acetoclastic and Hydrogenotrophic Methanogenesis under Mesophilic and Thermophilic Conditions

**DOI:** 10.3390/ijerph20043423

**Published:** 2023-02-15

**Authors:** Xiang Li, Yang Yang, Chen-Shun Lu, Takuro Kobayashi, Zhe Kong, Yong Hu

**Affiliations:** 1School of Environmental Science and Engineering, Nanjing Tech University, Nanjing 211816, China; 2Material Cycles Division, National Institute for Environmental Studies, 16-2 Onogawa, Tsukuba 305-8506, Japan; 3School of Environmental Science and Engineering, Suzhou University of Science and Technology, Suzhou 215009, China

**Keywords:** oleate, acetoclastic methanogenesis, hydrogenotrophic methanogenesis, temperature

## Abstract

This study investigated oleate inhibition concentration on mesophilic and thermophilic sludge by utilizing acetate and H_2_/CO_2_ (80:20, *v*/*v*) as substrate, respectively. In addition, another batch experiment was carried out to explore the influence of oleate loads (mM-oleate/g-VS) on methane production. Generally, the mesophilic anaerobic system was more stable than the thermophilic system, which embodied higher microbial abundance, higher methane yield, and higher oleate tolerance. Furthermore, this study provides a possible methanogenic pathway impacted by oleate under mesophilic and thermophilic conditions according to functional microbial composition. Lastly, this paper provides noticeable and avoidable oleate concentrations and loads under different experimental conditions as a guide for future anaerobic bioreactors of lipidic waste biodegradation.

## 1. Introduction

The amount of lipidic waste, such as fat, oil and grease (FOG), increases yearly due to population growth and catering industry development; waste oil alone was approximately 200 million tons worldwide specifically in 2020 [1]. Therefore, lipidic waste needs appropriate management which has been widely proposed [2]. Traditional incineration technology not only consumes approximately 70 kWh-electricity/t-lipidic-waste annually, but also converts 20% weight of lipidic waste into fly ashes which further require landfills [3]. As a sustainable and eco-friendly biological technology, anaerobic digestion (AD) has been applied to lipidic waste biodegradation because FOG presents a high chemical oxygen demand/total organic carbon (COD/TOC) ratio, which is eligible to serve as an ideal co-digestion substrate [4]. In addition, it has been reported that FOG can enhance gaseous renewable energy source production, such as methane, by 250~350% [4,5]. However, as a critical step in FOG biodegradation, long chain fatty acids (LCFAs) have long β-oxidation pathways, which become the rate-limiting step in the AD process [6]. Microscopically, LCFAs can adsorb to the cell membrane of anaerobic microorganisms and then generate block layers to obstruct cell membrane, which can prevent microbes from obtaining essential nutrients for their vital movements [7]. Simultaneously, LCFAs can poison anaerobic microbes and destroy their homeostasis such as pH, leading to damages such as transformation and death to microbes [8]. Macroscopically, sludge flotation and washout are the primary jeopardies of LCFA accumulation, and subsequently, the anaerobic bioreactor deteriorates due to sludge acidification [5,9,10]. Therefore, LCFAs have been widely recognized as inhibitors of the AD system. However, there are few pronounced opinions on the specific inhibitory concentration for anaerobic reactors due to the different experimental conditions used by researchers.

Some researchers attribute LCFAs inhibitory degree to molecular structure. Generally, LCFAs can be divided into unsaturated LCFAs and saturated LCFAs, in which unsaturated LCFAs are theoretically more obstructive than saturated ones due to their faster accumulation on microbial membranes [11]. Therefore, one research compared unsaturated oleate, saturated stearate and palmitate; the authors concluded that 1 mM-oleate or more than 4 mM-stearate and palmitate can inhibit 50% of the methanogenic activity of *Methanobacterium formicicum* [12]. On the other hand, Silva and Salvador discovered that 0.5 mM oleate can cause a decrease of over 50% of biogas production of two typical aetate-utilizing methane-producing archaea (MPA), while 2 mM palmitate can inhibit 50% of methane production by utilizing sodium acetate as substrate under mesophilic conditions [13]. Furthermore, some researchers ascribe the inhibitor degree to temperature. Previous studies have demonstrated that the IC_50_ of oleate is 0.35~0.79 mM at 55 °C and 2.35~4.30 mM at 30 °C, respectively, by utilizing acetate as substrate [14]. As temperature rises from a mesophilic condition to a thermophilic one, methanogenesis inhibition is more severe because of the increase of LCFA solubility [15]. In addition, some researchers consider that sludge types are possible factors; Hwe and Lettinga concluded that the IC_50_ of flocculent sludge is 0.53 mM, while granular sludge is 1.75 mM. These conclusions demonstrate that flocculent sludge is more sensitive to LCFAs than granular ones due to more sufficient contact area [14]. Some researchers take methane-producing archaea species into consideration. For example, Sousa and Salvador utilized 16S rRNA identification to detect two hydrogen-utilizing MPA (*Methanobacterium formicicum* and *Methanospirillum hungatei*) and two acetate-utilizing MPA (*Methanosaeta concilii* and *Methanosarcina mazei*), and proved that hydrogen-utilizing ones are more recalcitrant in a 1 mM oleate incubation [12]. Meanwhile, a previous study proved that a higher sludge-loading rate mediates hydrolytic conversion efficiency of anaerobic sludge and makes better use of substrate, but the MPA is apt to be poisoned on account of more LCFA accumulation [16]. Therefore, LCFA loads (mM-LCFA/g-biomass) are utilized to further manifest the relationship between LCFA concentration and sludge-loading rate. However, the relationship between LCFA load and specific methanogenic activity (SMA) results is still ambiguous. From the above discussion, we can observe that the conclusions are partial due to the unitary experimental conditions. Specifically, there are few systematical studies focusing on the influence of LCFA concentration and loads on methanogenic activity under different experimental conditions.

As a typical unsaturated LCFA, oleate (C_18:1_) has continuously attracted much attention for its abundance in wastewater [17]. In this paper, we utilized oleate as a representative LCFA in anaerobic serum vial bottles. First, we analyzed the initial microbial community composition of mesophilic and thermophilic seed sludge by 16S rRNA identification to further verify subsequent conclusions. Then, we carried out SMA tests by utilizing sodium acetate or H_2_/CO_2_ (80:20, *v*/*v*), respectively, as substrate under mesophilic or thermophilic environments to investigate methanogenesis inhibition and then analyze oleate IC_50_ under these conditions. Lastly, we carried out another batch experiment to explore the influence of oleate loads on methanogenic activity under acetoclastic or hydrogenotrophic pathways by utilizing initial mesophilic and thermophilic sludge.

## 2. Materials and Methods

### 2.1. 16S rRNA Sequencing Analysis

The microbial community of initial mesophilic and thermophilic sludge was analyzed by 16S rRNA cloning and sequencing. The procedure consisted of microorganism DNA extraction, primers selection for polymerase chain reaction (PCR) amplification, electrophoretic separation and fragment sequencing. The detailed procedure about genomic sampling and analysis was carried out according to a previous thesis [18].

### 2.2. Seed Sludge and Substrate

Sufficient mesophilic and thermophilic flocculent seed sludge was derived from a continuous-stirred tank reactor (CSTR) for lipidic waste biodegradation [19]. Subsequently, mesophilic seed sludge was cultivated in a mesophilic incubator (35 ± 1 °C) and thermophilic seed sludge was cultivated in a thermophilic incubator (55 ± 1 °C) for three days, respectively, in order to recover their methanogenesis activity. In the liquid phase, sodium acetate was utilized to cultivate acetate-utilizing MPA. The concentration of sodium acetate was 2000 mg-COD/L. A ratio of H_2_/CO_2_ (80:20, *v*/*v*) was utilized to cultivate hydrogen-utilizing archaea. Moderate phosphate butter was utilized to maintain the stability of all serum vial bottles and ensure sufficient absorption of substrate to microorganisms. The detailed concentration of various elements in the phosphate buffer are listed in Appendix A.

### 2.3. Experimental Design

Firstly, seed sludge, phosphate buffer, sodium oleate and distilled water were added in each anaerobic serum vial bottle according to calculation. Then, hydrochloric acid was utilized to adjust pH values to approximately 7. Subsequently, all serum vial bottles were compressed with a butyl stopper and then sealed with an aluminum crimp cap. The headspace of each bottle was flushed with N_2_ for 2 min to replace air when utilizing acetate as substrate. Next, sodium acetate was injected to cultivate acetate-utilizing MPA. Conversely, H_2_/CO_2_ (80:20, *v/v*) was used to fill the headspace when cultivating hydrogen-utilizing MPA. Finally, 250 mg/L Na_2_S·9H_2_O was injected into all bottles to ensure anaerobic environments. The total volume of each bottle was 120 mL and the liquid volume containing anaerobic seed sludge, phosphate buffer, substrate, and sodium oleate was maintained at 50 mL in all serum vial bottles. Subsequently, all of these bottles were statically cultivated at mesophilic (35 ± 1 °C) and thermophilic (55 ± 1 °C) conditions, respectively, in a shaker (MMS-220, EYELA) for heating and shaking. Each experiment was performed in three parallel vial bottles. Table 1 lists the detailed experimental conditions of diverse oleate concentration and sludge volatile solid contents, respectively. Gas production was measured three times a day until gas production verged to zero at a certain duration. This indicated that total substrate COD was almost converted to theoretical biogas yield. The biogas yield and composition were measured after sampling. Specifically, the duration of acetoclastic SMA in mesophilic sludge was about 7 days. On the other hand, the duration of hydrogenotrophic SMA in mesophilic and thermophilic sludge was about 3 and 4 days, respectively.

### 2.4. Analytical Methods

Sludge volatile solid content was detected according to standard methods (American Public Health Association, 2012). The COD was determined by COD digestion apparatus and matching COD measuring instrument (HACH, Loveland, CO, USA). Biogas production was measured by glass syringe. The pH value was measured by a handheld pH meter (LAQUAtwin, Horiba Ltd., Albany, NY, USA). The composition of biogas was determined by a gas chromatography (Shimadzu GC 2014) equipped with a flame ionization detector and a StabiliwaxR-DA capillary column (Resteck, Bellefonte, PA, USA). 

## 3. Results and Discussion

### 3.1. Microorganism Composition in Initial Mesophilic and Thermophilic Sludge

According to the 16S rRNA analysis, in mesophilic sludge, the two main archaea (over 100 OTU numbers) are *Methanosaeta* and *Methanospirillum* (as shown in Table 2), which accounts for 1928 (similarity 99.70%) and 1583 (similarity 99.96%), respectively. As the most abundant archaea in mesophilic sludge, acetoclastic *Methanosaeta* only utilize acetate as a carbon source to produce methane [20,21]. On the other hand, as a typical hydrogen-utilizing archaea, *Methanospirillum* is common in mesophilic anaerobic digestion systems [22]. Meanwhile, two dominant archaea in thermophilic sludge are *Methanoculleus* and *Methanothermobacter*. Specifically, *Methanoculleus* accounts for 1529 OTU with 99.92% similarity, while *Methanothermobacter* only accounts for 189 OTU with 99.99% similarity. These two archaea are both able to synthesize methane through hydrogenotrophic pathways [23,24,25]. On the other hand, there are few acetate-utilizing MPA in thermophilic seed sludge, which indicates that acetate-utilizing MPA have difficulty surviving under thermophilic conditions. Therefore, an acetoclastic SMA test under thermophilic conditions is not representative.

However, data for the main bacteria shown in Appendix A indicate that phylum *Thermotogae* was the maximum phylum in initial mesophilic and thermophilic sludge, which accounts for 6044 and 26573 OTU numbers, respectively. The optimum growing temperature of most species was from 45 °C to 80 °C, which explains their higher abundance in thermophilic sludge [26]. Furthermore, as species that play an important role in LCFA anaerobic biodegradation, abundant *Firmicutes* promote microbial tolerance to oleate in anaerobic environments [17,27]. Generally, the abundance of *Firmicute* is higher in initial mesophilic sludge than in thermophilic, which indicates that mesophilic sludge has stronger viability under lipid-rich conditions. On the other hand, the total *Bacteroidetes* out numbers in mesophilic and thermophilic sludge were 13970 and 5591, respectively, which suggests that *Bacteroidetes* biodiversity can decline due to temperature rises [28]. As an oleate sensitive species, they are not suitable to grow under the presence of oleate.

Other anaerobic bacteria in the phylum *Chloroflexi*, such as the mesophilic genus *T78*, can obtain energy by reductive dehalogenation of organic chlorinated compounds, which conforms to *T78* abundance in mesophilic sludge [29]. Furthermore, the genus *Anaerobaculum* is a common thermophilic acetogen in the phylum *Synergistetes,* while the genus *vadinCA02* and *HA73* are mesophilic; their functions hydrolyze complex organic matters into acetate, butyrate, lactate, H_2_ and CO_2_ [30]. On the other hand, *Actinomyces* plays an important role in hydrolyzation and biodegradation of organic matters into volatile fatty acids [31]. Therefore, these species came from the seed sludge from the previous experiment and showed little impact on subsequent experiments.

In general, Figure 1 visually presents that the abundance of microbial species in mesophilic sludge is much larger than that in thermophilic sludge, especially acetate-utilizing and hydrogen-utilizing archaea for methane production. Therefore, we predict that anaerobic mesophilic reactors used for lipidic waste degradation are more stable and reliable than thermophilic ones because they possess larger microbial quantity and more abundant microbial species, though the thermophilic condition promotes the substrate hydrolyzation process and theoretically has higher potential to produce methane [32].

### 3.2. Effect of Oleate Concentration on Specific Methanogenic Activity

On account of there being bare acetate-utilizing archaea in the initial thermophilic sludge presented in Section 3.1, we did not carry out acetoclastic SMA tests in thermophilic sludge. Figure 2 presents the results of the SMA test under different experimental conditions. Broadly speaking, with the increase of oleate concentration, their hindrance to MPA was more and more remarkable. This result agrees with the consensus that in lipid-rich anaerobic biodegradation reactors, typical LCFAs, such as oleate, have low biodegradation rates, low bioconversion rates and low bioavailability [8,33]. Therefore, LCFAs have been recognized as typical inhibitors in AD systems. On the other hand, biogas production verged to zero, especially when oleate concentrations were over 1 mM for acetate-utilizing MPA and 3 mM for hydrogen-utilizing ones, which provides an avoidable oleate concentration for future anaerobic reactor research.

According to the comparison in Figure 2a,b, oleate inhibition on acetate-utilizing MPAs is more severe than on hydrogen-utilizing ones under the same temperature. This result indicates that hydrogen has higher bioavailability than acetate which can be ascribed to molecule mass. Specifically, hydrogen has a smaller molecular structure than acetate, which is more conducive for anaerobic microbes to carry out mass transportation [34]. Furthermore, according to the comparison in Figure 2b,c, at low oleate concentrations, higher temperatures aggravate oleate inhibition on selected seed sludge. This can be attributed to the fact that higher temperatures accelerate LCFA solubility, which further aggravates the quantity of biogas bioconversion [15]. Overall, oleate IC_50_ was 0.42 mM, 1.57 mM, and 1.66 mM, as shown in Figure 2a to Figure 2c, respectively. This result further manifests critical oleate concentration in SMA research under different experimental conditions, and it provides a noteworthy oleate concentration for future anaerobic digestion systems.

At the microbial level, the most abundant bacteria *Thermotogae* have enormous potential to convert organic matter, such as acetate, into hydrogen, and require acetate as a feedstock. This function can be a reasonable explanation for the result that hydrogenotrophic SMA showed larger oleate IC_50_ than acetoclastic SMA, as presented in Figure 2. On the other hand, the hydrogenotrophic oleate IC_50_ was greater than the acetoclastic under the same temperature. This result is similar to the conclusion that acetate-utilizing archaea have an inferior capacity to survive under lipid-rich AD reactors [12]. Although the number of acetate-utilizing archaea was slightly larger than the hydrogen-utilizing ones, molecular size seemed to be a more significant factor than microbial abundance, resulting in lower IC_50_ of acetoclastic SMA than hydrogenotrophic SMA. Based on the above reasons, the acetate-utilizing capacity of anaerobic archaea only accounted for approximately one third of the hydrogen in the mesophilic sludge in the presence of sodium oleate.

Previous studies have investigated oleate inhibition degree and oleate IC_50_ under different experimental conditions. Table 3 lists the research results. According to the comparison with previous studies, the mesophilic acetoclastic oleate IC_50_ result was 0.42 mM in our research, which is in agreement with 0.5 mM in a previous study. This result can be contributed to the same predominant acetate-utilizing archaea *Methanosaeta*. However, in mesophilic hydrogenotrophic anaerobic serum vial bottles, oleate IC_50_ was far more than 1 mM in our study. This can be ascribed to diverse predominant hydrogen-utilizing MPA such as *Methanobacterium formicicum*. Therefore, we predict that *Methanospirillum* shows more tolerance to LCFAs than pure *Methanobacterium formicicum* [12]. In future lipidic waste biodegradation anaerobic reactors, the predominant archaea in seed sludge should be investigated to assess their tolerance on LCFA and their influence on reactor stability.

### 3.3. Effect of Oleate Loads on Specific Methanogenic Activity

According to the analysis of oleate IC_50_ presented in Section 3.2, the oleate inhibition concentration in serum vial bottles was maintained at 0.5 mM for acetoclastic batch assays and 1.5 mM for hydrogenotrophic batch assays, respectively, to further explore the influence of sludge loading rate on methanogenic activity. However, according to the conventional Monod equation, which shows a functional relationship between microbial specific growth rate and fundamental substrate concentrations, we adjusted typical SMA units to g-COD/d rather than conventional g-COD/g-VS/d due to the change of sludge volatile solid contents in different experimental conditions. This can scientifically eliminate the effect of microbial quantity on SMA results [35,36]. Figure 3 presents modified SMA results of diverse sludge loading rates in mesophilic and thermophilic sludge.

In general, since total numbers of archaea in thermophilic sludge were smaller than that in mesophilic sludge, this factor may cause the decrease of biogas production under the same sludge loading rate. On the other hand, whether LCFAs exist or not, mesophilic sludge consumes more substrate at a certain stage than thermophilic sludge under the same sludge loading rate. This result can be ascribed to mesophilic sludge having stronger adaptability and stability than thermophilic sludge [37]. Meanwhile, as temperature rises from a mesophilic condition to a thermophilic condition, sodium oleate surface tension promptly decreases [38]. This indicates that the superficial area of sodium oleate in thermophilic sludge is larger than in mesophilic, thus thermophilic sodium oleate shows better mass transfer capacity than mesophilic and theoretically has superior bioavailability. However, though the thermophilic sludge hydrolysis rate coefficient is theoretically larger than that of mesophilic sludge, increased intermediate toxicity of sodium oleate will prejudice methane production in lipid-rich AD systems [32]. This further leads to worse and worse oleate availability in thermophilic sludge as the sludge loading rate increases. Therefore, mesophilic anaerobic bioreactors for lipidic biodegradation are more stable than thermophilic ones and can be more widely applied for actual lipid-rich wastewater [39].

In addition, as shown in Figure 3a,b, methane production was almost in line with sludge loading rate in mesophilic sludge. This result indicates that increased anaerobic microbial quantity promotes methanogenic activity below 10 g/L sludge loading rate, and higher sludge loading rates need further study to demonstrate their possible negative effects on anaerobic bioreactors, such as sludge acidification. In hydrogen-utilizing serum vial bottles, the presence of oleate deeply impeded the methanogenesis process from 2 to 4 g/L sludge loading rate, which indicates that low microbial quantity is apt to be inhibited by LCFAs in anaerobic reactors. Then, oleate showed low interference on SMA results as sludge loading rate increased to 10 g/L sludge loading rate, which can be attributed to increased microbial abundance more adaptable to inhibitors and oleate can be utilized as a carbon source for microbial vital movement and methanogenesis [40]. Therefore, sludge loading rate is a crucial index in anaerobic thermophilic digestion of lipidic waste biodegradation which needs comprehensive consideration.

Oleate loads (mM-oleate/g-VS) can be utilized to further manifest the relationship between oleate concentration and sludge loading rate. Figure 4 presents the relationship between oleate loads and SMA results. As shown in Figure 4b, in general, the SMA result (%) decreased as the oleate load increased. On the other hand, the noticeable oleate load of hydrogenotrophic SMA in mesophilic sludge was 6.68 mM-oleate/g-VS. However, the results of acetoclastic SMA in mesophilic sludge and hydrogenotrophic SMA in thermophilic sludge show a primary slow increase and a subsequent sharp decrease. The optimal oleate load for acetoclastic SMA in mesophilic sludge and hydrogenotrophic SMA in thermophilic sludge is 5 mM-oleate/g-VS for both. This indicates that a moderate oleate load is not only slightly detrimental to anaerobic microbes, but also has bioavailable potential. This result agrees with the conclusion that LCFAs theoretically have high methane production potential (1010 mL CH_4_/g VS), though specific methane production potential is not similar [41,42]. Meanwhile, a previous study proved that LCFA-rich waste can be utilized as co-substrate in anaerobic digesters and the bioreactor showed favorable stability [43]. Therefore, in future FOG biodegradation anaerobic reactors, we should not only take the adverse effect of oleate loads on anaerobic sludge into consideration so that we can maintain the stability of the AD reactor as much as possible, but we should also consider their biogas production potential to take full advantage of their benefits.

Figure 5 presents the possible methanogenesis pathway in mesophilic and thermophilic sludge. In general, in mesophilic anaerobic serum bottles, acetate and hydrogen are utilized by abundant *Methanosaeta* and *Methanospirillum,* respectively. However, oleate can adhere to archaea cytomembrane or form sediments with ions which impede archaea from carrying out the utilization of substrate and the methanogenesis process. *Firmicutes* migrate around oleate to guarantee microbial tolerance on toxic organisms entirely. On the other hand, the only remaining methane production process is carried out by hydrogen-utilizing archaea such as *Methanoculleus* in thermophilic sludge, which indicates that fewer types of substrates are available for thermophilic bioreactors and this may become the reason of instability in thermophilic bioreactors. On the other hand, as a carbon source for MPA, hydrogen shows larger utilization efficiency and higher biogas conversion potential than acetate. This shows that gaseous substrates have higher mass transfer potential in batch assays. However, in actual AD plants, the concentration of gaseous substrate is hard to maintain at a high level, and H_2_ is also dangerous when operating, so the actual application of gaseous substrates needs further study.

From the above mentioned, the thermophilic sludge showed negative oleate tolerance and utilization capacity compared to the mesophilic sludge due to lower bacteria and archaea varieties and quantities in the initial sludge, which indicates that oleate concentration is a more critical index in a thermophilic AD system compared to a mesophilic AD system. Meanwhile, the oleate load is also a crucial factor in an AD bioreactor because it reveals oleate tolerance and adaptation under diverse microbial abundance. Generally, LCFAs such as oleate not only show negative effects on reactor stability and microbial methanogenesis, but also show diverse biogas production capacities under different sludge loading rates. Therefore, oleate load in lipid-rich AD reactors deserves systematical research. In the future, studies on anaerobic bioreactors for lipidic waste biodegradation, oleate concentration and load in thermophilic reactors deserve more attention. Meanwhile, this paper provides a notable oleate concentration and loads for future study.

## 4. Conclusions

Generally, in the presence of oleate, the acetate-utilizing capacity of anaerobic archaea only accounts for approximately one third of hydrogen in mesophilic sludge on account of the mutual impact of archaea abundance and molecular structure. On the other hand, oleate concentration is a critical index in anaerobic bioreactors because oleate shows inhibition on methanogenesis under hydrogenotrophic pathways in mesophilic sludge. Meanwhile, 16S rRNA analysis manifests that the abundance of initial archaea and other bacteria decrease as temperature rises, which is an unstable sign for thermophilic AD systems for lipidic waste treatment. However, oleate load deserves more attention because a moderate oleate load (5 mM-oleate/g-VS) promotes methanogenesis activity in mesophilic sludge. Therefore, notable and avoidable oleate concentrations and loads are provided in this study as a guide for future anaerobic lipidic waste digestion bioreactors.

## Figures and Tables

**Figure 1 ijerph-20-03423-f001:**
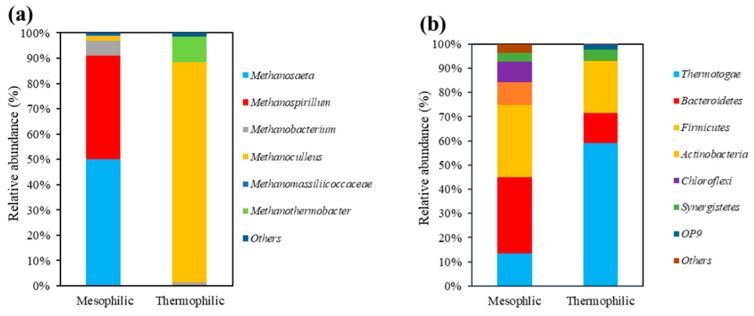
Microorganism composition in mesophilic and thermophilic sludge: (**a**) archaea genus composition, (**b**) bacteria phylum composition.

**Figure 2 ijerph-20-03423-f002:**
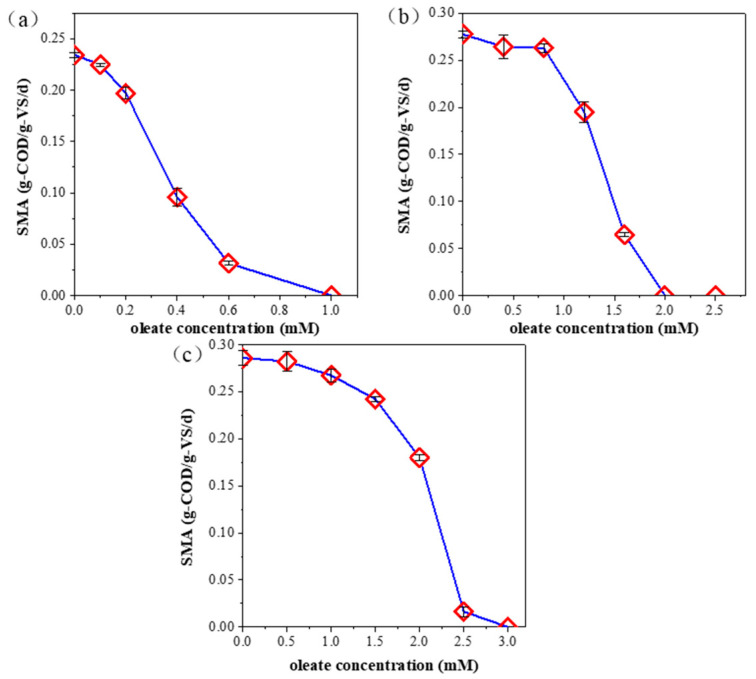
Effect of oleate concentration on SMA results: (**a**) acetoclastic SMA in mesophilic sludge, (**b**) hydrogenotrophic SMA in mesophilic sludge, (**c**) hydrogenotrophic SMA in thermophilic sludge.

**Figure 3 ijerph-20-03423-f003:**
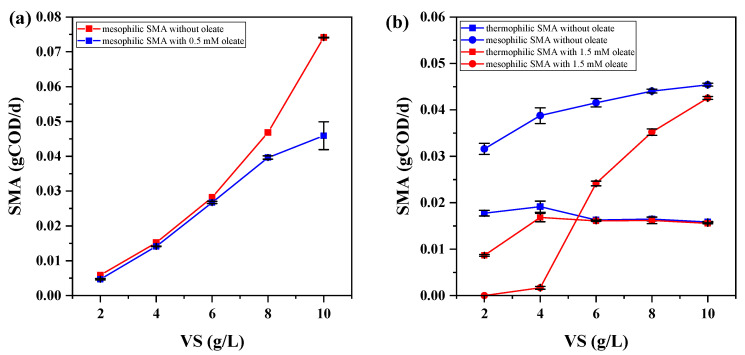
Effect of sludge loading rate on SMA results: (**a**) acetoclastic SMA in mesophilic sludge, (**b**) hydrogenotrophic SMA in mesophilic and thermophilic sludge.

**Figure 4 ijerph-20-03423-f004:**
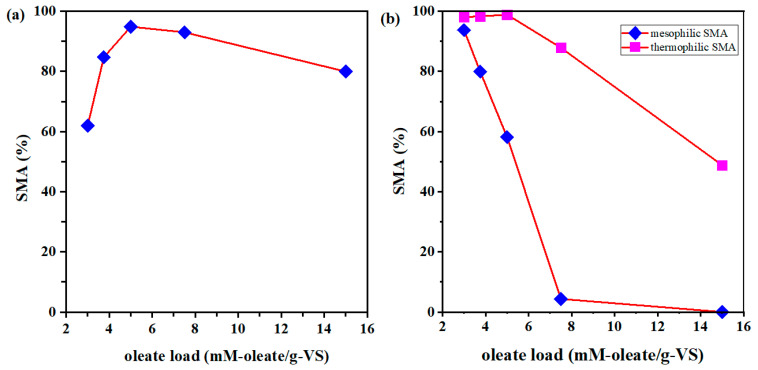
The relationship between oleate loads and SMA: (**a**) acetoclastic SMA in mesophilic sludge, (**b**) hydrogenotrophic SMA in mesophilic and thermophilic sludge.

**Figure 5 ijerph-20-03423-f005:**
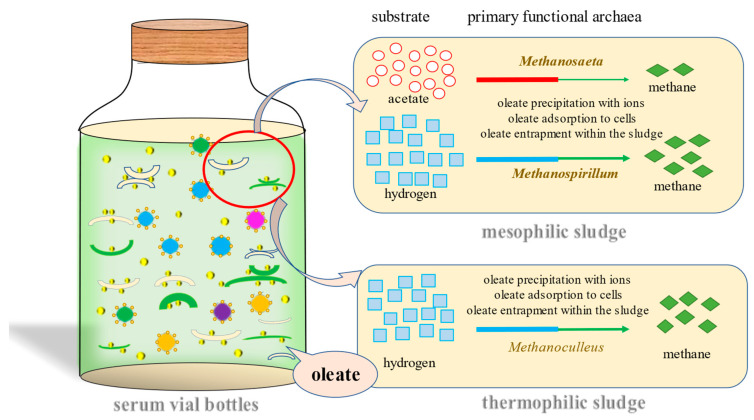
Possible methane conversion pathway impacted by oleate in mesophilic and thermophilic sludge.

**Table 1 ijerph-20-03423-t001:** Detailed experimental conditions.

Number	VS (g/L)	Oleate Concentration (mM)	Number	VS (g/L)	Oleate Concentration (mM)
MA1	4	0	MA2	4	0.1
MA3	4	0.2	MA4	4	0.4
MA5	4	0.6	MA6	4	1.0
MH1	4	0	TH1	4	0
MH2	4	0.4	TH2	4	0.5
MH3	4	0.8	TH3	4	1.0
MH4	4	1.2	TH4	4	1.5
MH5	4	1.6	TH5	4	2.0
MH6	4	2.0	TH6	4	2.5
MH7	4	2.5	TH7	4	3.0
MA7	2	0	MA8	2	0.5
MA9	4	0	MA10	4	0.5
MA11	6	0	MA12	6	0.5
MA13	8	0	MA14	8	0.5
MA15	10	0	MA16	10	0.5
MH8	2	0	TH8	2	0
MH9	2	1.5	TH9	2	1.5
MH10	4	0	TH10	4	0
MH11	4	1.5	TH11	4	1.5
MH12	6	0	TH12	6	0
MH13	6	1.5	TH13	6	1.5
MH14	8	0	TH14	8	0
MH15	8	1.5	TH15	8	1.5
MH16	10	0	TH16	10	0
MH17	10	1.5	TH17	10	1.5

MA represents serum vial bottles utilizing acetate as substrate under mesophilic conditions. MH represents serum vial bottles utilizing hydrogen as substrate under mesophilic conditions. TH represents serum vial bottles utilizing hydrogen as substrate under thermophilic conditions.

**Table 2 ijerph-20-03423-t002:** Primary archaea composition in mesophilic and thermophilic sludge.

Temperature	Phylum	Genus	Number of OTU	Percentage (%)	Similarity (%)
Mesophilic	*Euryarchaeota*	*Methanosaeta*	1928	50.00	99.70
		*Methanospirillum*	1583	41.05	99.96
		*Methanobacterium*	226	5.86	
		*Methanoculleus*	72	1.87	
		*Methanomassiliicoccaceae*	26	0.67	
		*Others*	21	0.54	
	Total		3856	100.00	
Thermophilic	*Euryarchaeota*	*Methanoculleus*	1529	86.92	99.92
		*Methanothermobacter*	179	10.18	99.99
		*Methanobacterium*	25	1.42	
		*Others*	26	1.48	
	Total		1759	100.00	

**Table 3 ijerph-20-03423-t003:** Oleate IC_50_ under different experimental conditions.

Temperature	Substrate	Dominant Archaea	IC_50_ (mM)	Ref.
mesophilic	acetate	*Methanosaeta*	0.42	this study
mesophilic	hydrogen	*Methanospirillum*	1.57	this study
thermophilic	hydrogen	*Methanoculleus*	1.66	this study
mesophilic	acetate	*Methanosaeta* and *Methanosarcina*	<0.50	[13]
mesophilic	hydrogen	*Methanobacterium*	1.00	[12]
mesophilic	hydrogen	*Methanospirillum*	0.30	[12]
mesophilic	acetate		0.35	[14]
thermophilic	acetate		0.53	[14]

## Data Availability

The research data can be provided if other researchers need them and research data will not provide here.

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
