# Peer review of "Oleate Impacts on Acetoclastic and Hydrogenotrophic Methanogenesis under Mesophilic and Thermophilic Conditions"

_ijerph, 2023, doi:10.3390/ijerph20043423_

Round 1

Reviewer 1 Report

The authors of the paper carried out a study of the anaerobic digestion of oleate as a representative compound of long chain fatty acids present in wastewater. Batch experiments were performed using acetate and mixtures of hydrogen and carbon dioxide separately, using different concentrations of oleate.

The subject is interesting and some contributions are presented that may help in the understanding of the mechanisms of LCFA inhibition in anaerobic digestion.

In the materials and methods section it is necessary to clarify whether the seed sludge is flocculent or granular and then in the results section to discuss the possible effect of inoculum type on the performance of anaerobic digestion in the presence of oleate.

In section 2.3 the duration of the kinetics should be included and adequately justified by considering methane production rates over time.

Change the adjective "terrible" in line 273 to technical terminology.

Change "swilled" to "flushed".

Reviewer 2 Report

Journal: IJERPH

COMMENTS

The paper could be interesting but some improvements are proposed:

o   Page 2, line 88. There is a mistake in the title section.

o   Section 2.1. More information with the fundamental of the section because it can make more simple the reading and the understanding of the paper. If the reader needs more information, the original paper could be consulted.

o   Lines 98-101. This information should be given in a table

o   Section 2.3. More information about the experimental design has to be provided. Examples:

§  What was the volume of the reactors?

§  What was the proportion of inoculum? Was the inoculum acclimated to the experimental conditions?

§  How was made the headspace filling? Time? How do the authors make sure that the H2 is accessible to the microorganisms in the liquid?

§  Picture and scheme of the experimental system would be appreciated.

§  What is about the TA condition (reactors utilizing acetate in thermophilic)? Why did not the authors try to compare to mesophilic?

§  What were the exact temperatures in mesophilic and thermophilic assays?

o   Section 2.4. Not all the methods were described, for example: VS, COD, etc… Please revise the overall document and complete the section.

o   Some acronyms are not described in the text. For example: OTU. Revise this point please.

o   Results and discussion section: From my point of view, this part could be redrafted to make clearer the obtained results and the discussion of previous references.

o   Revision of English style would be desirable, because sometimes is really difficult the understanding of the text. I would appreciate the overall revision, but specific examples could be found in lines 94, 96, 115, 116, 182, 187, etc.

o   Please, check the reference format

Round 2

Reviewer 1 Report

The authors performed all the suggested revisions.

Reviewer 2 Report

The paper could be accepted.